# HDAC Inhibitors: Therapeutic Potential in Fibrosis-Associated Human Diseases

**DOI:** 10.3390/ijms20061329

**Published:** 2019-03-16

**Authors:** Somy Yoon, Gaeun Kang, Gwang Hyeon Eom

**Affiliations:** 1Department of Pharmacology, Chonnam National University Medical School, Hwasun 58128, Korea; oouate@naver.com; 2Division of Clinical Pharmacology, Chonnam National University Hospital, Gwangju 61469, Korea

**Keywords:** fibrosis, HDAC, HDAC inhibitor, therapeutics

## Abstract

Fibrosis is characterized by excessive deposition of the extracellular matrix and develops because of fibroblast differentiation during the process of inflammation. Various cytokines stimulate resident fibroblasts, which differentiate into myofibroblasts. Myofibroblasts actively synthesize an excessive amount of extracellular matrix, which indicates pathologic fibrosis. Although initial fibrosis is a physiologic response, the accumulated fibrous material causes failure of normal organ function. Cardiac fibrosis interferes with proper diastole, whereas pulmonary fibrosis results in chronic hypoxia; liver cirrhosis induces portal hypertension, and overgrowth of fibroblasts in the conjunctiva is a major cause of glaucoma surgical failure. Recently, several reports have clearly demonstrated the functional relevance of certain types of histone deacetylases (HDACs) in various kinds of fibrosis and the successful alleviation of the condition in animal models using HDAC inhibitors. In this review, we discuss the therapeutic potential of HDAC inhibitors in fibrosis-associated human diseases using results obtained from animal models.

## 1. Introduction

### 1.1. Fibrosis

Fibrosis is a type of reactive process characterized by excessive accumulation of fibrous connective material in tissues or organs [1]. When tissues or organs are injured, a fibroma is formed during the healing process [2], through a series of processes called scarring. Though fibrosis can sometimes be resolved spontaneously [3], the most common types of fibrosis are tightly linked with pathologic states [2].

Fibrosis is initiated by stimulated fibroblasts, and circulating fibrocytes also contribute minimally [4]. Transforming growth factor (TGF)-β is the most well established pro-fibrotic signal [5], and is mainly secreted by macrophages responding to inflammation in injured tissues [6]. Other notable factors include tumor necrosis factor (TNF)-α [7], platelet-derived growth factor (PDGF) [8], basic fibroblast growth factor (bFGF) [9], and connective tissue growth factor (CTGF) [10]. These stimulants provoke fibroblast differentiation into myofibroblasts, which exacerbates extracellular matrix deposition [11]. The molecular pathway for fibroblast activation, SMAD phosphorylation, and subsequent SMAD nuclear translocation is well established [12]. The PI3K-AKT-mTOR signal cascade also contributes to fibroblast activation [13].

During fibrosis, epithelial–mesenchymal transition (EMT), a type of transdifferentiation of epithelial cells, is also an important step. Among the numerous intracellular regulators, the roles of SNAILs, basic helix-loop-helix (bHLH), and zinc-finger E box binding (ZEB) are well established in transdifferentiation of epithelial cells [14]. In terms of induction, TGF-β strongly promotes EMT. TGF-β causes transdifferentiation of epithelial cells predominantly through SMAD family signaling; however, PI3K-AKT-mTOR and RHOA pathways are also activated in response to TGF-β stimuli [14]. The specific mechanism of EMT is quite similar to fibroblast differentiation.

### 1.2. HDAC and HDAC Inhibitors

Histone deacetylases remove the acetyl moiety from histone tails [15]. Posttranslational modification of histone tails regulates transcriptional activity by modulating chromatin compaction [16]. Histone acetylation neutralizes the positive charge of lysine, which results in weakened binding of histones with DNA, resulting in less compacted DNA. On the other hand, histone deacetylation induces chromatin compaction. Removal of the acetyl group results in the tight association of the positively charged lysine with the negatively charged DNA. Hence, transcriptional activity is suppressed by histone deacetylation. Histone acetylation is mediated by histone acetyltransferases (HATs), whereas histone deacetylation is carried out by histone deacetylases (HDACs). HATs and HDACs finely regulate the histone acetylation status and thereby transcription.

Eighteen HDACs have been identified in mammals and are divided into four classes. HDAC1, -2, -3, and -8 are class I HDACs. HDAC4, -5, -6, -7, -9, and -10 are class II HDACs. HDAC6 and -10 contain two copies of the catalytic site. Recently, class II HDACs have been subgrouped as class IIa (HDAC4, -5, -7, and -9) and class IIb (HDAC6 and -10). The Sirtuin family (Sirt1-7) are classified as class III HDAC. HDAC11 is the only member of class IV HDAC. Class I, II, and IV HDACs require zinc ions to deacetylase their substrate and share a conserved functional deacetylation domain [17], suggesting that a single compound could inhibit all zinc-dependent HDACs simultaneously. Unlike zinc-dependent HDACs, sirtuins require NAD^+^ to execute deacetylation. Specifically, class III HDACs can be suppressed by nicotinamides.

### 1.3. Functional Relevance of HDAC in Fibrogenesis

Previous reports have independently delineated the role of HDACs in the development of fibrosis. Even though the specific mechanism of HDAC is somewhat different, cumulative evidence indicates that HDACs accelerate fibrogenesis in a redundant manner and that HDAC inhibitors (HDACIs) successfully regulate fibrosis. We briefly summarize the therapeutic potential of HDACIs in fibrosis in Figure 1.

According to HDACI studies, HDACs function as pro-inflammatory molecules that trigger secretion of pro-fibrotic cytokines [18]. HDACI interferes with expression and/or secretion of interleukin (IL)-1β [19], IL-6 (a master regulator in inflammation) [20,21], and TNF-α [22]. Zhu et al. observed that active HDAC3 specifically recruits NF-κB/p65 and thereby regulates TNF-α production in response to lipopolysaccharide stimulation [22]. In the next steps, various subtypes of HDACs are significantly associated with the inflammation process. In interferon gamma stimulated cells, HDACs accumulate in the promoter region and provoke the expression of genes required for the inflammatory response [23]. Specifically, HDAC1/2 and HDAC3 were identified as inflammatory regulators in epithelial cells [17] and in fibroblast-like synoviocytes [24], respectively. Increased expression of HDACs stimulates fibroblast differentiation into myofibroblasts [25]. During chronic inflammation, various cytokines from inflammatory sites stimulate myofibroblast differentiation, which indicates that anti-inflammatory properties could also regulate fibrosis in an indirect manner. However, many previous reports show evidence supporting the direct regulation by HDACs in myofibroblast differentiation [25,26]. To overcome the indirect effect of HDACI in vivo, fibroblasts were isolated and cultured in vitro. HDACI significantly reduced myofibroblast differentiation triggered by TGF-β. HDACs also incited extracellular matrix formation [27]. HDAC1 and HDAC2, as components of Sin3A, epigenetically blocked collagen synthesis in a synchronous manner [28]. At least at particular points, HDACs contribute in various ways in each step of fibrosis. Taken together, these results support the concept that inhibition of HDACs is important for inhibiting the progression of fibrosis.

### 1.4. HDAC Inhibitors and Their Therapeutic Potential

HDAC was first identified as a novel teratogenic factor [29]. Class I HDACs regulate cell proliferation in a redundant manner. HDACs are closely linked with tumorigenic features such as proliferation, distant metastasis, and aneuploidy [29], and, generally, increased expression of HDACs is linked with poor prognosis [30,31]. Hence, pharmacologic inhibition of HDACs could be a potential strategy in the development of cancer treatment. To date, at least four HDAC inhibitors (HDACIs) have been approved by the United States Food and Drug Administration (US FDA): Vorinostat (2006, Zolinza^®^), romidepsin (2009, Istodax^®^), belinostat (2014, Beleodaq^®^), and panobinostat (2015, Farydak^®^). An HDACI was approved for hematologic malignancy [15,29]. Vorinostat and romidepsin were approved for cutaneous T cell lymphoma (CTCL) and peripheral T cell lymphoma (PTCL). Belinostat was also approved for relapsing PTCL. Panobinostat was licensed for multiple myeloma. The overall remission rate with these HDACIs ranges about 20–30%. However, long-term survival benefits are limited [29]. HDACIs have been approved and highlighted as an emerging option for anticancer regimens. Massive clinical trials have been undertaken to expand the clinical indication of approved HDACIs or even de novo inhibitors for solid tumors. However, the overall survival benefits are quite limited [32]. Many research groups have delineated the role of HDACs in various human diseases as well as the beneficial effects of HDACIs in the animal models of those diseases. Beyond showing anticancer properties, HDACIs successfully ameliorated the progression of atherosclerosis [33], myocardial death by ischemia–reperfusion injury [34], Alzheimer’s disease development [35], inflammation [18], and fibrosis-associated diseases.

Fibrosis itself is a physiological reaction; however, deposition of fibrous material interferes with the normal functioning of organs or tissues [1]. Hence, proper control of fibrosis is important to maintain physiological organ functions. In the clinic, limited options are available to control progression of fibrosis-associated diseases. For example, glucocorticoids or tyrosine kinase inhibitors are prescribed for pulmonary fibrosis, but the therapeutic outcomes are still limited [36]. Most patients suffer from progressive deterioration of pulmonary function despite conventional treatment regimens. Developing novel therapeutics to alleviate fibrosis is an urgent medical issue. In this review, we summarize the possible benefits of HDACIs as novel regulators of tissue fibrosis. We briefly summarize the fibrosis-associated human diseases in Table 1.

## 2. Experimental Outcomes of HDAC Inhibitors in Animal Models of Fibrosis-Associated Disease

### 2.1. Liver Cirrhosis

Liver cirrhosis involves chronic irreversible changes of the hepatic parenchyma to scar tissue, i.e., the process of fibrosis. Notable causes of liver cirrhosis include chronic alcohol consumption [53,54], non-alcoholic fatty liver disease [55], aflatoxin [56], and hepatitis virus infection [57]. Among the pathophysiologies of cirrhosis, chronic inflammation of the liver is a common underlying condition. Park et al. demonstrated an improvement of severity in liver cirrhosis and in the survival rate by the use of HDACIs [37]. Liver fibrosis was induced by bile duct ligation (BDL) in rats followed by administration of HDACIs. Activation of hepatic stellate cells (HSCs), the major source of hepatic myofibroblasts, was dramatically reduced in the HDACI-treatment group. HDACI arrested the cell cycle and even induced apoptosis in HSCs. HDACI ameliorated the hepatic dysfunction exacerbated by BDL and markedly improved the survival rate. Notably, the final outcome of HDACI treatment is superior to that of cyclooxygenase inhibition, indicating that HDACIs exert complex effects, including anti-inflammatory effects [29,37].

### 2.2. Cardiac Fibrosis

Accumulation of interstitial fibrosis in the heart aggravates cardiac dysfunction. The main function of the heart is supplying nutrition and oxygen to the peripheral tissues. Regular beating is thus mandatory for appropriate circulation. To control the series of sequential contractions, the heart has its own regulatory system controlled by an electronic drive generated in the sinoatrial (SA) node. Before contraction, the ventricle has to relax sufficiently to secure inflow of blood. In other words, efficient ventricular relaxation, or diastole, wherein the ventricular chamber is filled with blood, is important for effective pumping out, or systole. Cardiac fibrosis occurs in several conditions and results in secondary problems as follows.

Fibrosis in the atria disturbs normal conduction from the SA node [58]. Frequently, fibrotic foci generate additional autonomic signals, which are occasionally conducted to the ventricle resulting in arrhythmia [59,60]. Atrial arrhythmia itself results in both turbulence and stasis of blood and finally induces thrombosis, an important cause of cerebral infarction [61]. Furthermore, conduction of irregular beats to the ventricle is a notable cause of heart failure [62]. Fibrous material can also accumulate in the ventricle.

Cardiac hypertrophy is the major underlying mechanism of ventricular fibrosis [42,63,64,65,66]. Although cardiac hypertrophy is a kind of adaptive process to counter increased hemodynamics, chronic uncontrolled stimuli exacerbate microinflammation and myofibroblast differentiation. Fibrous changes in the ventricle might contribute to contractile force. However, they yield negative outcomes in the diastole phase, which is referred to as diastolic dysfunction [67]. Fibrosis-induced ventricular stiffness results in the failure of appropriate ventricular relaxation and decreases the ventricular blood volume. If diastolic dysfunction is not controlled, it can proceed to diastolic heart failure [68]. No effective drugs are available for diastolic heart failure. All clinical trials with beta blockers [69], angiotensin converting enzyme (ACE) inhibitors/angiotensin receptor blockers (ARB) [70,71], or aldosterone antagonists [72] failed to show improved survival rates in diastolic heart failure patients [73].

HDACIs can successfully control both atrial fibrosis and ventricular fibrosis. Liu et al. clearly demonstrated that trichostatin A (TSA), a pan-HDAC inhibitor, alleviated atrial fibrosis and subsequent atrial fibrillation (AF) [44]. In addition, TSA normalized connexin 40 remodeling. HDAC inhibition reversed conduction abnormalities and atrial automaticity. Seki et al. developed a canine model for atrial arrhythmia and atrial fibrosis [45]. In addition to TSA, class I HDAC specific inhibitors were tested. Both pan-HDACIs and class I HDACIs ameliorated atrial fibrosis and AF. Overall cardiac function was also improved in the HDACI-administered group. Inflammatory cell infiltration was markedly reduced.

A genetic ablation study of HDAC provided direct evidence to understand the role of HDAC in arrhythmia. Montgomery et al. deleted both HDAC1 and HDAC2 and found that the L-type and T-type calcium subunits were dysregulated [66]. Cacna1h and Cacna2d2 were markedly increased in HDAC1 and HDAC2 double knockout mice and the mice presented with fatal cardiac arrhythmia. Meraviglia et al. treated primary rat cardiomyocyte cultures with HDACI and suberanilohydroxamic acid (SAHA), and measured calcium current [74]. SAHA-treatment in cardiomyocytes ameliorated intracellular calcium handling and contractile performance through acetylation of the sarcoplasmic reticulum protein calcium ATPase 2. Taken together, these data suggest class I HDACs play a pivotal role in the development of atrial fibrosis and atrial arrhythmia.

Both right and left ventricular fibrosis was well controlled by HDACIs; however, few studies have revealed the role of HDACs in right ventricular fibrosis. Cho et al. induced right ventricular hypertrophy with monocrotaline or by pulmonary artery constriction [43]. Unlike captopril, an ACE inhibitor, HDACI significantly suppressed the fibrotic changes in the right ventricle. There is limited evidence on right ventricular hypertrophy and fibrosis, and more detailed studies are required. Left ventricular hypertrophy and fibrosis have been repeatedly reported by various research groups [40,41,42,63,64,75,76]. Pan-HDACI or selective class I HDACI alleviate the development of hypertrophy and progression of fibrosis [40,41,42]. Long-term treatment with HDACI showed remarkable improvement in heart failure transition and cardiac fibrosis in a rodent model [41,64]. HDACIs are notable therapeutics for diastolic heart failure patients with cardiac fibrosis as the major underlying cause [77]. HDACIs ameliorated both hypertrophy and cardiac fibrosis simultaneously. Hence, it is not clear whether the anti-fibrotic effect of HDACI is a direct effect or an indirect outcome of improving hypertrophy. To answer this question, Yoon et al. isolated and cultured cardiac fibroblasts from adult mice and observed that myofibroblast differentiation by TGF-b is attenuated when HDAC2 is inhibited in vitro [76]. Taking these data together, we conclude that HDACIs can directly regulate cardiac fibrosis.

### 2.3. Pulmonary Fibrosis

Pulmonary fibrosis is characterized by a chronic irreversible decline in pulmonary function. Typical symptoms include dry cough, shortness of breath, and limitation of exercise [78]. When the cause of pulmonary fibrosis is not determined, patients are diagnosed with idiopathic pulmonary fibrosis (IPF) [79]. Air pollution, cigarette smoking, and viral infection are regarded as causes of pulmonary fibrosis. Thus far, effective regimens to block the progression of pulmonary fibrosis remain limited [80]. The common pathophysiology of pulmonary fibrosis includes chronic inflammation in the lung parenchyma and deposition of fibrous extracellular matrix, which finally results in thickening of the alveolar wall. In the case of a known underlying cause, the primary cause of disease should be controlled adequately. However, there is no reported cause of IPF, and inhibition of fibrosis is the only therapeutic strategy [80,81].

Notable studies have reported the therapeutic potential of HDACIs in IPF. Guo et al. utilized normal human lung fibroblasts (NHLF) and induced fibrosis using TGF-β [25]. TSA abrogated NHLF differentiation into myofibroblasts and small interfering RNA against HDAC4 blocked smooth muscle alpha actin accumulation. In a separate result, Coward et al. demonstrated that epigenetic abnormalities in cyclooxygenase-2 expression were restored by HDAC inhibition, which induces resistance to pulmonary fibrogenesis [47].

### 2.4. Renal Fibrosis

Renal fibrosis is also determined by aberrant growth of residential fibroblasts and accumulation of excess fibrous materials in the renal parenchyma. Renal fibrosis is often associated with glomerulonephritis [10,82], focal segmental glomerulosclerosis [83], IgA nephropathy [84], and diabetic nephropathy [85]. Commonly, loss of glomeruli and substitution of fibrotic foci is observed [86]. For the molecular signaling cascade, the roles of TGF-β/SMAD and signal transducer and activator of transcription (STAT) 3 are well established [86,87]. The rodent model for renal fibrosis induced by unilateral ureteral obstruction (UUO) is widely used [88]. Pan-HDACIs, such as TSA [48,49,89] or CG200745 [51], have been shown to exhibit a renoprotective effect. HDACIs alleviated glomerular destruction and aberrant expansion of interstitial fibrosis. Similar to the results obtained with the in vivo administration of HDACIs, STAT3 signal was inhibited in vitro by TSA in NRK49F, a rat kidney fibroblast cell-line [89]. Beside directly inhibiting fibroblast differentiation, HDACIs also regulate epithelial–mesenchymal transition. Noh et al. injected streptozotocin (STZ) to induce diabetes in rats and treated them with TSA. The HDACI potently decreased fibrotic changes in STZ-treated rats. To visualize the direct effect of TSA on fibrosis, a rat kidney epithelial cell line, NRK52E, was also utilized. Treatment with TGF-β in the culture medium successfully induced fibrosis in NRK52E cells, and TSA effectively prevented these changes. Through a series of studies, HDAC2 was indicated as a master regulator in renal fibrosis [52].

### 2.5. Miscellaneous Diseases

Polycystic kidney disease (PKD) is an inherited disease whose main characteristics include progressive growth of cysts [90]. In addition to the kidney, cyst formation simultaneously occurs in different organs such as the liver, pancreas, or aortic root, which results in aneurysms [91]. A variety of genetic mutations have been indicated to be responsible factors in PKD. Polycystin-1 (PKD1), polycystin-2 (PKD2), and GANAB (PKD3) [92,93,94] are also indicated. Chronic interstitial inflammation and consequent fibrosis is a major feature of PKD. Because PKD mainly arises from genetic problems, no established therapeutics are available [90]. Ultimately, kidney transplantation would be required in end-stage renal disease patients. Several studies have applied animal models for human PKD and tested various chemicals to suppress disease progression [95,96,97]. Pkd1 mutation induced robust upregulation of HDAC6, which induced cystic fibrosis transmembrane conductance regulator (CFTR) and fibroblast cell proliferation. Tubacin, an HDAC6-specific inhibitor, attenuated cyst formation in a rodent model. Renal function was also improved in the tubacin-treated group. Thus, an HDAC6-targeted approach could be considered as a therapeutic strategy for PKD in the future [95].

Cystic fibrosis is a hereditary disease that generates cysts of varying sizes in most parts of the body including the liver, lung, kidney, and intestines [98]. Life-threatening comorbidities includes repeated-pulmonary infection and difficulty in breathing [99]. Mutation in cystic fibrosis transmembrane conductance regulator (CFTR) is a major pathophysiology of this fatal genetic disorder. Specific deletion of phenylalanine 508 (∆F508) results in unconventional folding and removal of CFTR, total loss of which results in failure to maintain the osmotic gradient [98]. Several studies have revealed that HDACIs can be considered as a therapeutic target for cystic fibrosis [100,101]. Recently, Bodas et al. utilized a transient overexpression system in HEK293 cells with ∆F508-CFTR and tested several HDAC inhibition approaches such as SAHA, Tubacin, and HDAC6-shRNA [100]. HDACIs successfully restored the intracellular trafficking that was abolished by misfolding of ∆F508-CFTR. In the presence of HDACIs, proteosomal degradation due to unconventional accumulation of ∆F508-CFTR in endoplasmic reticulum was markedly reduced. Furthermore, chronic treatment of HDACIs restored ion current malfunction of ∆F508-CFTR [102]. Possible mechanism, how HDACIs specifically ameliorates channel function defect of ∆F508-CFTR was further supported by another report. Pankow et al. cultured bronchial epithelial cells and evaluated the functional complex of ∆F508-CFTR by use of SAHA treated cell. ∆F508-CFTR formed different complex from wild type CFTR, SAHA exerted interactome remodeling of ∆F508-CFTR. Finally, SAHA-exposed human primary bronchial epithelial cells carrying ∆F508-CFTR was able to successfully regulate osmosis [101]. Thus, targeting HDACI could be a management strategy for cystic fibrosis patients.

During surgical procedures, skin fibroblasts are easily activated and are deeply associated with scarring [103]. Most of the scar formation by fibrosis is not harmful but is a cosmetic problem. However, fibrosis in the eye may restrict vision or interfere with drainage of aqueous humor and hence might be fatal [104]. Trabeculectomy surgery is considered an option to release the increased ocular pressure [105]. The main purpose of trabeculectomy is to generate a drainage outflow track for the aqueous humor [105]. Scar formation around the outflow track halts drainage, indicating surgical failure. Hence, anti-fibrotic management is important to improve surgical outcomes. Eye drops with steroids are widely used after trabeculectomy intervention; however, fatal side effects of topical steroids have been reported, including wound infection, hypotony, and paradoxical elevation of intraocular pressure [106,107]. Sung et al. administered TSA in a rat trabeculectomy model and compared the results with the clinical outcome of the steroid administered group. Topical TSA dramatically reduced fibrosis and vascularity, as effectively as steroids, without any apparent corneal epithelial toxicity. HDACI could thus be an alternative modality to steroids after trabeculectomy surgery [108].

Hypertrophic scars, or keloids, are a fibrosis-associated disorder [103]. It is difficult to manipulate because surgical procedures or similar intervention could cause reaggravation. Despite a limited number of reports, HDACI could be a novel therapeutic option for keloid or hypertrophic scars. Fitzgerald et al. observed upregulation of HDAC2 in keloid scars [109]. Furthermore, Russell et al. found that epigenetic alteration in keloid fibroblasts was normalized after treatment with TSA [110]. Diao et al. injected TSA in hypertrophic scars and observed regression of a pre-established scar [111]. Taken together, these data indicate that topical administration of HDACI could be an effective method to control the overgrowth of skin fibroblasts.

## 3. Limitations and Future Perspectives

Multiple HDACs are involved in various kinds of human disease. Sometimes, more than two HDACs play an opposite role in the development of a single disease, suggesting that pan-HDACIs can result in unwanted effects. HDAC2, a class I HDAC, induces cardiac hypertrophy, whereas class IIa HDACs potently suppress that response [75]. To minimize adverse drug reactions, subtype specific inhibitors should be developed. HDAC1 and HDAC2, class I HDACs, are highly expressed in most cells, and molecular homology between HDAC1 and HDAC2 is quite high [112,113]. Furthermore, class I, II, and IV HDACs share a conserved catalytic domain, indicating that it is difficult to develop a subtype-specific inhibitor [15,16,29]. However, few HDACIs are available for subtype specific HDAC inhibition. Only HDAC6, a class IIb HDAC, can be specifically blocked by a single compound [114]. Otherwise, more than two HDACs are affected simultaneously.

Besides histones, numerous “non-histone” proteins undergo acetylation dynamics. For this reason, HDACIs simultaneously induce histone compaction by modulating histones as well as affecting enzyme activity by non-histone protein acetylation, which indicates that HDAC inhibition might contribute to fatal side effects. Our group already reported the toxic side effects of an HDACI in a vulnerable subject in vascular calcification. Thus, HDACI fatally accelerates calcium deposition [115]. Hence, indirect inhibition should be considered to bypass these issues. Our group treated cultured cardiac fibroblasts with an Hsp70 inhibitor to mimic the inactivation of HDAC2. Hsp70 inhibition effectively attenuated myofibroblast differentiation as did HDACI [76]. Use of an indirect method to suppress a certain type of HDAC after understanding the detailed pathophysiology could overcome the adverse drug reaction to HDACIs without loss of efficacy.

## 4. Conclusions

In this review, we briefly summarize multiple types of animal studies regarding human diseases that share chronic inflammation and tissue fibrosis as underlying mechanisms. Tissue fibrosis results in interference with normal organ function, and no therapeutics are currently available for the condition. Therefore, anti-fibrotic agents need to be urgently developed for increasing both quality of life and survival rate of patients with fibrosis and related conditions. Numerous reports have repeatedly demonstrated the therapeutic potential of HDACIs in animal models of fibrosis-associated human disease. Hence, to create additional therapeutic options, it may be worth considering clinical trials for these diseases using FDA-approved HDACI.

## Figures and Tables

**Figure 1 ijms-20-01329-f001:**
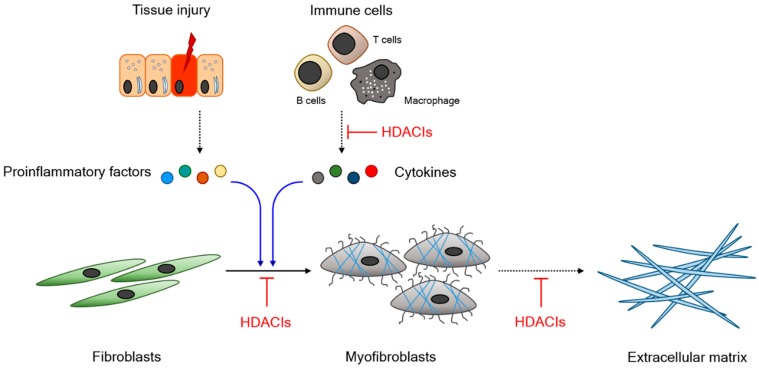
Schematic demonstration of the anti-fibrotic property of HDACIs. Injured tissue or activated immune cells secrete profibrotic factors, which induce fibroblast differentiation into myofibroblasts. Myofibroblasts actively synthesize extracellular matrix. HDACIs negatively regulate fibrosis. Dashed arrow: secretion; Blue arrow: stimulation; Black arrow: differentiation; Red blunted line: inhibition. Abbreviation; HDACI, Histone deacetylase inhibitor.

**Table 1 ijms-20-01329-t001:** HDACIs tested in animal or cellular studies for human fibrosis-associated diseases.

	HDAC Inhibitor	Selectivity	Model	Output (Except Fibrosis)	Reference
Liver cirrhosis	SAHA HNHA	pan pan	Bile duct ligation	Improved hepatic function Survival ↑	[37]
	MC1568	HDAC4/5/6	CCl_4_	HSC activation ↓	[38]
	Valproate	pan	thioacetamide	HSC activation ↓	[39]
Cardiac fibrosis	TSA SK7041	pan class I	Pressure overload	Heart failure ↓ Cardiac hypertrophy ↓	[40]
	Api-D	class I	Pressure overload	Heart failure ↓ Cardiac hypertrophy ↓	[41]
	TSA Scriptaid	pan pan	Pressure overload	Heart failure ↓ Cardiac hypertrophy ↓	[42]
	Valproate	pan	Pressure overload MCT	RV hypertrophy	[43]
	TSA	pan	TgHopX	Cx40 ↑ Normalized conduction	[44]
	Tacedinaline	class I	TgHopX pacing (dog)	Atrial fibrillation ↓ Immune cell infiltration ↓	[45]
Lung fibrosis	TSA	pan	TGF-β (NHLF cell)	Myofibroblast differentiation ↓	[25]
	SAHA	pan	Bleomycin	Lung compliance ↑ Airway resistance ↓	[46]
	SAHA panobinostat	pan pan	Primary cells from IPF patient	Correction of epigenetic abnormality	[47]
Renal fibrosis	TSA	pan	UUO	Immune cell infiltration ↓	[48]
	TSA	pan	UUO	Tubular cell apoptosis ↓	[49]
	Valproate	pan	UUO	Macrophage infiltration ↓	[50]
	CG200745	pan	UUO	Serum NGAL level ↓	[51]
	TSA Valproate SK7041	pan pan class I	STZ	Urine protein/Cr ↓ EMT ↓	[52]

Abbreviations: Api-D, apicidin derivative; CCl_4_, carbon tetrachloride; Cr, creatinine; EMT, epithelial-mesenchymal transition; HNHA, *N*-hydroxy-7-(2-naphthylthio) heptanomide; HSC, hepatic stellate cells; IPF, idiopathic pulmonary fibrosis; MCT, monocrotaline; NGAL, neutrophil gelatinase-associated lipocalin; NHLF, normal human lung fibroblast; pan, pan-HDAC inhibitor; RV, right ventricle; SAHA, suberoylanilide hydroxamic acid; SK7041, 3-(4-substituted phenyl)-*N*-hydroxy-2-propenamide; STZ, streptozotocin; TGF-β, transforming growth factor beta; TgHopX, transgenic mice expressing HopX; TSA, trichostatin A; UUO, unilateral ureteric obstruction. ↑, increase; ↓, decrease.

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
