# Peer review of "HDAC Inhibitors: Therapeutic Potential in Fibrosis-Associated Human Diseases"

_ijms, 2019, doi:10.3390/ijms20061329_

Round 1

Reviewer 1 Report

This review is a comprehensive compilation of studies carried out so far on the function of HDACs and the use of their inhibitors in fibrosis-associated diseases. The manuscript is well-organized and easy to follow, however an extensive corrections regarding English language is required.

Here below some minor issues to be addressed:

- Line 58: HDAC1 and not HDAC11

- More details are required for the legend of figure 1. e.g. the meaning of the arrows and blunted lines and specify the type of immune cells cited in the figure.

- In chapter 1.3 HDACs are described as proinflammatory molecules, however their inhibitor reduces the action of TGF-b (line 90) that is anti-inflammatory cytokine. this is contradictory.

- What is the relationship between fibrosis and cancer in chapter 1.4?

- The last sentence of page 5 (line 185) does not make sens.

- Line 205: Please define the cell line NRK52E. The effect of TGF-b on the induction of fibrosis in this cell line is mediated by which experimental mean? by TGF-b over-expression?

- Line 230: Please introduce the abbreviation SAHA.

Author Response

Response to Reviewers

We appreciate you giving us the opportunity to revise our manuscript. The main issues raised by the reviewers are (1) English editing by reviewer #1, (2) detailed descriptions of specific mechanisms by reviewer #2 and #3, and (3) correction of errors by reviewer #1 and #2. We have addressed all the critiques raised by the reviewers and have revised the manuscript. We hope the manuscript can now be considered suitable for publication. Please let us know if we can provide any further information.

Reviewer #1

We appreciate the reviewer’s kind suggestions.

1. English revision

An extensive corrections regarding English language is required.

The revised manuscript has been reedited by a professional English language editor.

2. Detailed description

More details are required for the legend of figure 1.

Please define the cell line NRK52E

We have added an explanation about the information presented in Figure 1. (currently, line #78-81). A description of the cell line in question has been added (currently, line #247-248).

3. Contradictory descriptions in chapter 1.3 and 1.4

In chapter 1.3 HDACs are described as proinflammatory molecules, however their inhibitor reduces the action of TGF-b (line 90) that is anti-inflammatory cytokine. this is contradictory. What is the relationship between fibrosis and cancer in chapter 1.4?

We apologize for the confusing description. We have improved the detail in our discussion of the mechanism (currently, line #84-90). In addition, we agree with the point regarding cancer in section 1.4. We have reduced the description.

4. Abbreviation of SAHA

Please introduce the abbreviation SAHA.

During revision, we added a number of additional sentences. We now introduce the abbreviation of SAHA at the first appearance (currently, line #193)

5. Typos

Line 58, Line 185

We have fixed the typos throughout the manuscript.

Reviewer 2 Report

HDAC have emerged as targets to develop new active compounds having anti-oncogenic, anti-infective... activity. Here the author proposes to review the potential of HDACi as therapeutics in fibrosis associated human diseases. The topic is exciting but the review rather confused and lack some information essential for readers. I feel that most of the essential pieces of information are gathered in this review but the text and the plan of the paper need to be thoroughly reviewed and paper needs to be rewritten !!!

Main remarks

1. Introduction

All the introductive chapters from 1.1 to 1.3 does not give sufficient information to readers to get a clear image of the incidence of fibrosis in human disease and health and on the molecular processes involved in Fibrosis even on what medications are currently available to treat fibrosis. Concerning HDAC and HAT, what are the demonstration of their role in fibrosis? What is the molecular formula of the various inhibitor and what is their mode of action?

All information essential for readers to understand the importance of HDACi in fibrosis therapy is lacking !!!

2. Fibrosis in human disease

1.1 Liver cirrhosis

All citation refers to the rat as an animal model !!!!

1.2 Cardiac fibrosis

Here most of the references relate experiments performed on animal models !!!

3. Limitations and future perspective

This chapter is rather vague and I do not see the limitation and perspective clearly exposed in the light of fibrosis and the chemotherapeutic application of HDACi.

4. Conclusion

I agree with the terms « briefly summarized » and may be diluted essential information for readers.

Specific comments

Line 96 « Aberrant overexpression of certain types of HDACs are repeatedly » please provides references

Line 96 please define what is « aberrant overexpression »

Line 97: What is a forced expression?

Line 99 What is the mechanism underlying « HDAC induction of angiogenesis »

Line 99 to 103 this paragraph is rather vague and need to be more explicit.

Line 120 I suggest changing the title of this chapter because most if not all the references refer to the animal model and not to human diseases

Line 121 « improvement in severity » this is not good for therapy.  Maybe the authors wanted to express «  improvement in the fibrosis status ».

Author Response

Response to Reviewers

We appreciate you giving us the opportunity to revise our manuscript. The main issues raised by the reviewers are (1) English editing by reviewer #1, (2) detailed descriptions of specific mechanisms by reviewer #2 and #3, and (3) correction of errors by reviewer #1 and #2. We have addressed all the critiques raised by the reviewers and have revised the manuscript. We hope the manuscript can now be considered suitable for publication. Please let us know if we can provide any further information.

Reviewer #2

We thank reviewer #2 for the valuable comments.

1. Improve contents

All information essential for readers to understand the importance of HDACi in fibrosis therapy is lacking !!!

We have added considerable information to the revised manuscript and rearranged the chapters to improve the manuscript. The detailed mechanism underlying the action of HDACs in fibrogenesis has been added (currently, line #84-90), and the importance of HDACi in fibrosis therapy is discussed (currently, line #124-130).

2. Subheading

I suggest changing the title of this chapter because most if not all the references refer to the animal model and not to human diseases

We agree with reviewer’s comment. Since all content in the manuscript discusses animal models for human disease, we have changed the subheading of chapter 2. (currently, line #141)

3. Vague description

This chapter is rather vague and I do not see the limitation and perspective clearly exposed in the light of fibrosis and the chemotherapeutic application of HDACi.

Line 99 to 103 this paragraph is rather vague and need to be more explicit.

We have revised the chapter 3. More detailed descriptions have been added (currently, line #302-312).

According to the comment of reviewer #1, we reduced the description of HDAC and its role in cancer (previously, line #99-103). We removed several sentences and rewrote (currently, line #105-109).

4. Typo

Line 121 « improvement in severity » this is not good for therapy.  Maybe the authors wanted to express «  improvement in the fibrosis status ».

We have revised the text.

Reviewer 3 Report

In this review Yoon and coauthors give a general overview on HDAC and HDAC inhibitors role during fibrogenesis. Several organs are functionally compromised by aberrant overexpression of HDACs that promote cancerous features by regulating myofibroblasts differentiation from fibroblasts, by inducing angiogenesis or provoking distant metastasis. On the other hand, it is also demonstrated the anti-inflammatory action of HDACIs, that significantly counteract these aberrant cellular mechanisms by leading to beneficial effects.

The authors pointed out on multiple human diseases where inflammation and tissue fibrosis interfere with organ function, in particular they raised the need to develop anti-fibrotics aimed to increase a survival rate. From my point of view, although the review is well written and the main points of this intriguing problem have been mentioned, the authors should explain in more detail two aspects:

1)      acethylation and deacethylation molecular mechanisms have been extensively explained in 1.2 paragraph and the general final effect of HDACs (as pro-inflammatory factors) and HDACIs (as anti-inflammatory ones) have been cited. I think that the authors should explain in a more extended way how the previous molecular mechanisms result in those final effects considering that HDACIs  is the pro-transcriptional condition but used in a therapeutic anticancer regimens, aimed to minimize every synthesis process. The conceptual link is not so evident;

2)      the authors mention the HDAC inhibition also as potential reverse factor of cardiac conduction abnormality and atrial automaticity (line 154, page 5) and they insert the citation about connexin40 remodelling by TSA. It is also known that pro-fibrotic HDACs overexpression dysregulate cardiac ion channels such as calcium channels, Scn3b and Kcne1 thus leading to the long QT or Brugada syndrome (Montgomery et al, 2007; Monteforte et al, 2012). On the other hand it has been demonstrated that HDACIs has a crucial role in improving SERCA2 activity in cardiomyocyte calcium handling process (Meraviglia et al, 2018). I think these aspects should be considered in cardiac fibrosis of this review.

Author Response

Response to Reviewers

We appreciate you giving us the opportunity to revise our manuscript. The main issues raised by the reviewers are (1) English editing by reviewer #1, (2) detailed descriptions of specific mechanisms by reviewer #2 and #3, and (3) correction of errors by reviewer #1 and #2. We have addressed all the critiques raised by the reviewers and have revised the manuscript. We hope the manuscript can now be considered suitable for publication. Please let us know if we can provide any further information.

Reviewer #3

We appreciate reviewer #3 for the critical comments.

1. Detailed description

I think that the authors should explain in a more extended way how the previous molecular mechanisms result in those final effects considering that HDACIs is the pro-transcriptional condition but used in a therapeutic anticancer regimens, aimed to minimize every synthesis process. The conceptual link is not so evident;

We truly appreciate this comment. The description of the molecular mechanism underlying the action of HDAC in fibrogenesis was somewhat missed. We have added considerable text regarding this to the revised manuscript (currently, line #84-90, #124-130)

2. Arrhythmia and HDAC

It is also known that pro-fibrotic HDACs overexpression dysregulate cardiac ion channels such as calcium channels, Scn3b and Kcne1 thus leading to the long QT or Brugada syndrome (Montgomery et al, 2007; Monteforte et al, 2012). On the other hand it has been demonstrated that HDACIs has a crucial role in improving SERCA2 activity in cardiomyocyte calcium handling process (Meraviglia et al, 2018). I think these aspects should be considered in cardiac fibrosis of this review.

We intentionally omitted a detailed description of cardiac arrhythmia because most of this content has already been discussed in a previous review by our group (Yoon and Eom, CMJ, 2016). Our group previously demonstrated the dysregulation of Scn3b and Kcne1 by transient overexpression of HDAC2 in cardiomyocytes. However, we did not expand our preliminary observation. Hence, we would like to skip the discussion regarding Scn3b and Kcne1. We cordially request the reviewer to understand out point-of-view.

We have added Meraviglia’s findings to the revised manuscript (currently, line #189-196)

Round 2

Reviewer 2 Report

Thank you for taking into account the comments. I just have a problem with the short paragraph line 264 that describe the activity of HDACi against CFTR. To my knowledge mutation of the phenylalanine 508 abrogate (dysregulate) the capability of CFTR to act as an osmotic regulator. I do not understand how HDCi can restore this capability? Can the authors give more detail on the way HDACi act?

Author Response

We appreciate the reviewer's comment.

1. Detailed description

 Can the authors give more detail on the way HDACi act?

We agree with reviewer's comment. Still, detailed mechanism how HDACi ameliorates dF508-CFTR remains unclear. However, phenotypically HDACi significantly retarded protein degradation and restored ion current of dF508-CFTR. We have discussed possible mechanism of HDACi (currently, line #274~283)